# PAGE: Equilibrate Personalization and Generalization in Federated Learning

Submission Id: 1206

## ABSTRACT

Federated learning (FL) is becoming a major driving force behind machine learning as a service, where customers (clients) collaboratively benefit from shared local updates under the orchestration of the service provider (server). Representing clients' current demands and the server's future demand, local model personalization and global model generalization are separately investigated, as the ill-effects of *data heterogeneity* enforce the community to focus on one over the other. However, these two seemingly competing goals are of equal importance rather than black and white issues, and should be achieved simultaneously. In this paper, we propose the first algorithm to balance personalization and generalization on top of game theory, dubbed PAGE, which reshapes FL as a co-opetition game between clients and the server. To explore the equilibrium, PAGE further formulates the game as Markov decision processes, and leverages the reinforcement learning algorithm, which simplifies the solving complexity. Extensive experiments on four widespread datasets show that PAGE outperforms state-of-the-art FL baselines in terms of global and local prediction accuracy simultaneously, and the accuracy can be improved by up to 35.20% and 39.91%, respectively. In addition, biased variants of PAGE imply promising adaptiveness to demand shifts in practice.

## CCS CONCEPTS

• **Computing methodologies** → **Machine learning**; **Distributed computing methodologies**; Reinforcement learning; • **Human-centered computing** → **Ubiquitous and mobile computing**.

## KEYWORDS

Federated learning, personalization and generalization, game theory, reinforcement learning

**ACM Reference Format:**
Anonymous Author(s). 2024. PAGE: Equilibrate Personalization and Generalization in Federated Learning. In *Proceedings of the ACM Web Conference 2024 (WWW '24)*. ACM, New York, NY, USA, 12 pages. https://doi.org/XXXXXXX.XXXXXXX

## 1 INTRODUCTION

With the rapid proliferation of data constantly generated on pervasive mobile and Web-of-Things (WoT) devices, federated learning

(FL) has emerged as a promising distributed machine learning (ML) paradigm that enables efficient data usage by unleashing the computation power on devices [20, 38]. In typical FL (TFL) [1, 10, 19, 25, 31], represented by FedAvg [31], a central server orchestrates a group of clients to train a single global model with desirable generalization by iteratively averaging local models rather than accessing raw data. Serving as a step towards high prediction accuracy and efficiency for ML-as-a-service (MLaaS), TFL is poised to revolutionize myriad WoT applications, such as the next word prediction on Google's Gboard on Android [6], healthcare [28], and e-commerce [32], etc.

However, TFL suffers severely from the *data heterogeneity* [26] issue, which is a fundamental challenge attributed to non-independent identically distributed (Non-i.i.d.) local data. To be specific, the prediction accuracy of a single global model on individual clients is significantly reduced in the presence of heterogeneous local data distributions. For instance, clients from different demographics are likely to require totally different prediction results for the same sample due to diverse cultural nuances, while a single global model cannot generalize well in this case.

To overcome the ill-effects of *data heterogeneity*, personalized FL (PFL) has sparked increasing interest during the past few years, where customized local models are constructed for individual clients to provide satisfactory personalization [12, 24, 33, 45]. Currently, the research trend is to accommodate the generalized global model as personalized local models. In this case, global model generalization is inevitably sacrificed with the improvement of local model personalization [36]. (An in-depth discussion of more related works is given in Section 2 and Appendix A.) But it is tempting to ask: Is the personalized local model in PFL, or perhaps the generalized global model in TFL, the most practical demand on earth? Although this pair of seemingly competing goals has enforced the FL community to focus on one over the other, it is never a black and white issue. Taking MLaaS as an example, customers require local models with desirable personalization, which is a current demand. On the contrary, generalized global models are pursued by the service provider to yield a better initialization to fine-tune local models for numerous new participants, which is referred to as the future demand.

Recently, Chen *et al.* [9] tried to draw attention back from personalization to their reconciliation, where the optimization priority between personalization and generalization was eliminated. Besides, the widely used regularizer was proven less effective and hence removed. Still, a definite insight into the equality between personalization and generalization was not claimed. To extend their insight, we specify that personalization and generalization share equal status in FL, and the balance between them is much needed. Back to the MLaaS scenario, balance refers to a moderate condition satisfying current and future demands simultaneously. Yet, an intuitive question springs to mind:

*How to achieve the balance between local model personalization and global model generalization in FL?*

In response, we propose a personalization and generalization equilibrium (PAGE) FL algorithm. Following the optimization problem in [9], we formulate FL as a joint evolution with mutual restraints between global and local models by removing the regularizer. Such an evolution is intractable, as the optimization of competing objectives would get out of control with the removal of the regularizer. Intuitively, the iterative evolution can be viewed as a co-opetition game, where the personas of clients and the server switch to players with leader-follower relations. As a result, to balance the competing objectives, PAGE establishes an implicit relation between local and global models through a feedback multi-stage multi-leader single-follower (MLSF) Stackelberg game [4]. Additionally, to simplify the exploration of the game equilibrium, i,e., balance, PAGE further re-formulates the game as Markov decision processes (MDPs) [5], and leverages the deep deterministic policy gradient (DDPG) [27] algorithm.

The main **contributions** are summarized as follows:

- To the best of our knowledge, PAGE is the first algorithm to balance generalization and personalization in FL. In particular, PAGE establishes the relation between personalization and generalization on top of game theory.
- We re-formulate the game as server-level and client-level MDPs, and explore the equilibrium by reinforcement learning (RL). Through rigorous analysis, the existence of the equilibrium is proved.
- We evaluate PAGE on four widespread databases. Experimental results show that PAGE outperforms the state-of-the-art (SOTA) PFL and TFL in terms of global and local prediction accuracy simultaneously, and the accuracy can be improved by up to 35.20% and 39.91%, respectively. Besides, biased variants of PAGE imply promising adaptiveness to varying demand shifts in practice.

## 2 RELATED WORK

Since the birth of FL, *data heterogeneity* has been a root cause of the tension between generalization and personalization. Accordingly, the research community has been divided into TFL [1, 10, 19, 25, 31] and PTL [9, 12, 24, 33, 45], focusing on global model generalization and local model personalization, respectively. Below we discuss the SOTA baselines most relevant to PAGE, and a more comprehensive literature review can be found in Appendix A.

**Typical federated learning.** Solutions for *data heterogeneity* stemmed from FedAvg [31], which was a standard and fundamental algorithm. Shortly, it was proven hard to meet Non-i.i.d. data [26]. Later on, to mitigate this issue, Li *et al.* [25] proposed FedProx to generalize FedAvg by adding a proximal term to the objective, which improved the stability facing heterogeneous data. Similarly, FedDyn investigated linear and quadratic penalty terms [1]. Different from these regularization methods, SCAFFOLD corrected local updates through variance reduction [19]. Recently, Chen *et al.* [10] proposed Dap-FL to adaptively control local contributions for aggregation. Although the above TFL algorithms could yield expected global model generalization, the single global model setting struggled

to satisfy customers' current demands in MLaaS, i.e., local model personalization.

**Personalized federated learning.** To overcome *data heterogeneity*, PFL has drawn significant research interest in training customized models adapting to diverse local data. For instance, pFedMe optimized a bi-level problem using regularized local loss functions with $L_2$-norm, where personalized local models were decoupled from the global model optimization [12]. Ditto conducted a similar regularization method, but differed by switching the priority between global and local objectives [24]. Besides, Singhal *et al.* leveraged model-agnostic meta-learning to fine-tune local models [33]. Most recently, Zhang *et al.* [45] proposed FedALA, which adaptively aggregated the downloaded global model and local models towards local objectives at the element level. However, PFL fared less well in global model generalization, which cannot meet the future demand of service providers in practice. One closely relevant work was FED-ROD, where an implicit regularizer was introduced to consider generalization in the presence of personalization [9]. Although FED-ROD decoupled local and global models, a definite insight into their equal statuses was absent.

To the best of our knowledge, no prior arts take the balance of local model personalization and global model generalization into account, while PAGE bridges this gap through game theory, thereby satisfying current and future demands simultaneously. More importantly, PAGE converts sub-games into MDPs, and derives the equilibrium by adaptively adjusting local training hyper-parameters and aggregation weights on top of RL.

## 3 PROBLEM STATEMENT

In this section, we first formalize TFL and PFL systems, then identify the problem to be solved in this paper[1]. Generally, FL involves $N$ clients $\mathbb{C} = \{c_i, i = 1, \cdots, N\}$ and a central server $CS$. Each $c_i$ has a private local dataset $D_i = \{(x_{i,k}, y_{i,k}), k = 1, \cdots, |D_i|\}$, where $|D_i|$ is the data size, $x_{i,k}$ is the feature of a specific sample, and $y_{i,k}$ is the corresponding label. Also, $CS$ owns a public dataset $D_{CS}$. The goal of TFL and PFL is to collaboratively train global and local models, respectively. Supposing $f_i(x_{i,k}, y_{i,k}; w_i)$ denotes $c_i$'s local loss function (simply expressed as $f_i(w_i)$), the global loss function is denoted by $F(\cdot)$ and defined as:

$$F(W) = \sum_{i=1}^{N} \left( p_i \cdot \mathbb{E}_{D_i} \left[ f_i(x_{i,k}, y_{i,k}; w_i) \right] \right) = \sum_{i=1}^{N} \left( p_i \cdot f_i(w_i) \right), \quad (1)$$

where $w_i$ is $c_i$'s local model, $W$ is the global model, $p_i \in (0, 1)$ is the aggregation weight, and $\sum_{i=1}^{N} p_i = 1$.

Mathematically, TFL aims to train a single global model with promising generalization, shown as:

$$W^* = \underset{W}{\arg\min} \, F(D_1, \cdots, D_N; W), \quad (2)$$

where $W^*$ is the converged global model. At the opposite end of the spectrum, to tackle data heterogeneity issues, PFL customizes

---

[1]For clarity, we summarize important notations in Appendix B.

local models with satisfactory personalization, formally given as:

$$\begin{cases} W^* = \underset{W}{\arg\min} \left\{ F(W) := \sum_{i=1}^{N} (p_i \cdot (f_i(w_i) + \mathcal{R}_i)) \right\}, \\ \text{s.t. } w_i^* = \underset{w_i}{\arg\min} \left\{ f_i(w_i) + \mathcal{R}_i \right\}, i = 1, \cdots, N, \end{cases} \quad (3)$$

where $w_i^*$ is the optimal local model, and the regularizer $\mathcal{R}_i$ controls the strength of $W$ to $w_i$.

Different from TFL and PFL, we concentrate on balancing global model generalization and local model personalization, rather than facilitating any of them to a position of prominence. Following [9], we define an optimization problem:

$$\mathbf{P_0} : \begin{cases} W^* = \underset{W}{\arg\min} \left\{ F(W) := \sum_{i=1}^{N} (p_i \cdot f_i(w_i)) \right\}, \\ w_i^* = \underset{w_i}{\arg\min} \left\{ f_i(w_i) \right\}, i = 1, \cdots, N. \end{cases}$$

In this case, $CS$ and $c_i$ would conduct an iterative co-opetition, aiming at a joint evolution with mutual restraints between $W$ and $w_i$. Specifically, in any given round $t = 1, \cdots, T$, each $c_i$ initializes the local model $w_i(t)$ as the most recent global model $W(t)$ received from $CS$. Then, $c_i$ updates $w_i(t)$ for $\alpha_i(t)$ epochs, expressed as:

$$\hat{w}_i(t) = Train\left(\eta_i(t), \alpha_i(t); w_i(t)\right), \quad (4)$$

where $\hat{w}_i(t)$ is the updated local model, and $\eta_i(t)$ is the learning rate. Subsequently, each $c_i$ uploads $\hat{w}_i(t)$ to $CS$, and $CS$ assigns $p_i(t)$ for every $\hat{w}_i(t)$ to update the global model by aggregation, shown as:

$$W(t+1) = \sum_{i=1}^{N} \left(p_i(t) \cdot \hat{w}_i(t)\right). \quad (5)$$

## 4 PROPOSED METHOD: PAGE

### 4.1 Game (Relation) Establishment

To control the delicate balance in $\mathbf{P_0}$, it is essential to establish a more effective relation between $W$ and $w_i$. In general, the balance-controlling factors are equivalent to the counterparts impacting $f_i(\cdot)$ and $F(\cdot)$. Empirical results show that the most significant factors are $\alpha_i(t)$, $\eta_i(t)$, and $p_i(t)$ [10, 40]. Concretely, a larger (smaller) $\alpha_i(t)$ provides more (fewer) steps of the optimization of $f_i(\cdot)$, thereby contributing more (lesser) to local model fitness over $D_i$, i.e., local model personalization. $\eta_i(t)$ wields the influence in a similar way. Besides, $\alpha_i(t)$ and $\eta_i(t)$ impact $F(\cdot)$ in an indirect manner, where $f_i(\cdot)$ plays a role in a bridge. Loosely speaking, over-optimized $f_i(\cdot)$ derived from larger $\alpha_i(t)$ and/or $\eta_i(t)$ holds down the convergence of $F(\cdot)$ to some extent, i.e., excessive local model personalization deteriorates global model generalization. Yet, appropriate $p_i(t)$ could mitigate the bias of over-optimized $f_i(\cdot)$ to facilitate the convergence of $F(\cdot)$, which, in turn, drags $f_i(\cdot)$ from overfitting. More critically, the influence of these balance-controlling factors on either personalization or generalization might even go beyond the apparently positive or negative correlation in practice, which exacerbates the complexity of the relation establishment.

From a game theory point of view, the iterative evolution between $w_i(t)$ and $W(t)$ subject to balance-controlling factors can be regarded as a multi-stage co-opetition game between clients and $CS$ with leader-follower sequences, where leaders move ahead of the follower in each stage. On this ground, we re-formulate $\mathbf{P_0}$ as a

feedback multi-stage MLSF Stackelberg game in Definition 1, based on which an implicit relation between $W$ and $w_i$ is established.

**DEFINITION 1.** $\mathbf{P_0}$ can be formulated as a feedback multi-stage MLSF Stackelberg game, defined as:

$$\mathbf{P_0'} = \left[\!\!\left[\left\langle \{c_i\}_{i=1}^{N} \in \mathbb{C}, CS \right\rangle, \left\langle \{g_i(t) \in \mathcal{G}_i\}_{i=1}^{N}, g_{CS}(t) \in \mathcal{G}_{CS} \right\rangle, \right. \\ \left. \left\langle \{u_i(t)\}_{i=1}^{N}, u_{CS}(t) \right\rangle, z(t) \in \mathcal{Z}, t = 1, \cdots, T \right]\!\!\right], \text{ where}$$

- $c_i, i = 1, \cdots, N$ are leaders, and $CS$ is the follower.
- $t = 1, \cdots, T$ represents the stage of the game. Note that the initial global model distribution is not involved in $\mathbf{P_0'}$.
- $g_i(t) = [\alpha_i(t), \eta_i(t)]$ is $c_i$'s strategy in the $t$-th stage, and $\mathcal{G}_i$ is the strategy space.
- $g_{CS}(t) = [p_1(t), \cdots, p_N(t)]$ is $CS$'s reacting strategy to all $g_i(t), i = 1, \cdots, N$, and $\mathcal{G}_{CS}$ is the strategy space.
- $u_i(t) = 1/f_i(\hat{w}_i(t))$ is $c_i$'s utility function.
- $u_{CS}(t) = 1/F(W(t+1)) = 1/\sum_{i=1}^{N}(p_i(t) \cdot f_i(\hat{w}_i(t)))$ is $CS$'s utility function.
- $z(t)$ is the gaming condition, and $\mathcal{Z}$ is the condition space.

Definition 1 depicts dynamic conflict situations between clients and $CS$ over time, in which each $c_i$ operates $g_i(t)$, and $CS$ optimizes $g_{CS}(t)$ subject to the constraints of all clients' strategies in each stage. Also, clients are able to infer $CS$'s reaction to any strategies they operate. Therefore, each $c_i$ could operate a strategy that maximizes the utility, given the predicted behavior of $CS$.

Notably, the equilibrium of the game $\mathbf{P_0'}$ provides a terminating condition for the pursuing balance in $\mathbf{P_0}$, whose existence is confirmed at the end of this section (Theorem 1). Next, in line with the general equilibrium solving method in Stackelberg games [4], we split $\mathbf{P_0'}$ as the Server-level and Client-level sub-games to explore the appropriate strategy sequences in the equilibrium separately.

### 4.2 Strategy Exploration in the Server-level Sub-game

For the server-level sub-game, the equilibrium of $\mathbf{P_0'}$ indicates the optimal strategy sequence of $CS$, where the strategy in the current stage hinges on the gaming result in the previous stage and impacts next-stage strategies. However, the optimal strategy sequence is intractable through general backward induction algorithms [4], as the complexity increases exponentially with $t$.

Intuitively, such an over-time strategy conducting process is equivalent to an MDP [5], where $CS$ makes decisions about $p_i(t)$ sequentially through interacting with the environment, i.e., evaluating local updates. In other words, the MDP 3-tuple could be naturally found in the server-level sub-game, and the optimal strategy sequence could be solved by RL algorithms. Therefore, we first model the Server-level sub-game as an MDP $\langle \mathcal{S}_{CS}, \mathcal{A}_{CS}, R_{CS}(\cdot) \rangle$, where $\mathcal{S}_{CS}$ is the state space, $\mathcal{A}_{CS} \equiv \mathcal{G}_{CS}$ is the action space, and $R_{CS}(\cdot)$ is the reward function. Below we define the 3-tuple in detail.

- *State:* $s_{CS}(t) \triangleq [\hat{acc}_1(t), \cdots, \hat{acc}_N(t)] \in \mathcal{S}_{CS}$, where $\hat{acc}_i(t)$ is the prediction accuracy of $\hat{w}_i(t)$ on $D_{CS}$.
- *Action:* $a_{CS}(t) \triangleq g_{CS}(t) = [p_1(t), \cdots, p_N(t)] \in \mathcal{A}_{CS}$.
- *Reward:* $r_{CS}(t) = R_{CS}(s_{CS}(t), a_{CS}(t), s_{CS}(t+1)) \triangleq u_{CS}(t) = 1/\sum_{i=1}^{N}(p_i(t) \cdot f_i(\hat{w}_i(t)))$.

Mathematically, the Server-level MDP is defined as:

$$\mathbf{P}_0'\_CS \ : \ \max_{\mu_{CS}(\cdot)} J_{CS}(\cdot),$$

where $\mu_{CS}(\cdot) : s_{CS}(t) \to a_{CS}(t)$ is the policy, $J_{CS}(\cdot) = \sum_{t=1}^{T}(\gamma^{t-1} \cdot r_{CS}(t))$, and $\gamma$ is the discount factor. Note that $\mathbf{P}_0'\_CS$ is approximately equivalent to the Server-level sub-game, as $\gamma$ is usually set as 0.99 in practice.

Due to high-dimensional and continuous action and state space, we introduce DDPG, which consists of a MainNet and a Target-Net with the same *Actor-Critic* structure [27], to solve $\mathbf{P}_0'\_CS$. In the MainNet, the *Actor* is expressed as $\mu_{CS}\left(\cdot; \theta_{CS}^{\mu}(t)\right)$, which takes $s_{CS}(t)$ as the input and outputs $a_{CS}(t)$ through the parameterized policy $\theta_{CS}^{\mu}(t)$. The *Critic* takes $s_{CS}(t)$ and $a_{CS}(t)$ as the input and outputs the value of the parameterized state-action function $Q_{CS}(\cdot; \theta_{CS}^{Q}(t))$. In addition, the TargetNet is a copy of the Main-Net, which is parameterized by $\mu_{CS}'(\cdot; \theta_{CS}^{\mu'}(t))$ and $Q_{CS}^{\mu'}(\cdot; \theta_{CS}^{Q'}(t))$. The detailed strategy exploration process is shown in Algorithm 1, where the best policy $\mu_{CS}^{*}(\cdot)$ outputs the selected action sequence $A_{CS}^{*} = \left[a_{CS}^{*}(1), \cdots, a_{CS}^{*}(T)\right]$, which is $CS$'s optimal strategy sequence $G_{CS}^{*}(1) = \left[g_{CS}^{*}(1), \cdots, g_{CS}^{*}(T)\right] \equiv A_{CS}^{*}$ in the equilibrium of $\mathbf{P}_0'$.

---

**Algorithm 1** Global Aggregation Weights Tuning

---

**Input:** $l_{CS}^{Cri}$ and $l_{CS}^{Act}$ are the learning rates for *Critic* and *Actor* in the MainNet; $\beta_{CS}$ is a tiny updating rate for the TargetNet; $|B|$ is the batch size.

**Output:** $p_i(t)|i=1,\cdots,N, t=1,\cdots,T$.

1: Initialize $\theta_{CS}^{\mu}(\cdot), \theta_{CS}^{Q}(\cdot), \theta_{CS}^{\mu'}(\cdot)$, and $\theta_{CS}^{Q'}(\cdot)$;
2: **for** $t = 1, \cdots, T$ **do**
3:    Observe $s_{CS}(t)$, and hence calculate $r_{CS}(t)$;
4:    Randomly sample a batch of experience tuples
      $(s_{CS}(\xi), a_{CS}(\xi), r_{CS}(\xi), s_{CS}(\xi+1))$, $\xi=1,\cdots,|B|$;
5:    **for** $\xi = 1, \cdots, |B|$ **do**
6:       Calculate $y_{CS}(\xi) = r_{CS}(\xi) + \gamma \cdot Q_{CS}^{\mu'}(s_{CS}(\xi + 1), \mu_i'(s_{CS}(\xi + 1); \theta_{CS}^{\mu'}(t-1)); \theta_{CS}^{Q'}(t-1))$;
7:    **end for**
8:    Calculate $Loss_{CS}(t-1) = 1/|B| \sum_{\xi=1}^{|B|}(y_{CS}(\xi) - Q_{CS}^{\mu}(s_{CS}(\xi), a_{CS}(\xi); \theta_{CS}^{Q}(t-1)))^2$;
9:    Update $\theta_{CS}^{Q}(t), \theta_{CS}^{\mu}(t), \theta_{CS}^{\mu'}(t)$, and $\theta_{CS}^{Q'}(t)$ as follows:
      $\theta_{CS}^{Q}(t) = \theta_{CS}^{Q}(t-1) - l_{CS}^{Cri} \cdot \nabla_{\theta_{CS}^{Q}} Loss_{CS}(t-1),$
      $\theta_{CS}^{\mu}(t) = \theta_{CS}^{\mu}(t-1) + l_{CS}^{Act} \cdot \nabla_{\theta_{CS}^{\mu}} J_{CS}(t-1),$
      $\theta_{CS}^{\mu'}(t) = \beta_{CS} \cdot \theta_{CS}^{\mu}(t-1) + (1-\beta_{CS}) \cdot \theta_{CS}^{\mu'}(t-1),$
      $\theta_{CS}^{Q'}(t) = \beta_{CS} \cdot \theta_{CS}^{Q}(t-1) + (1-\beta_{CS}) \cdot \theta_{CS}^{Q'}(t-1);$
10:  **end for**
11: **return** $\mu_{CS}^{*}(\cdot)$;
12: **return** $G_{CS}^{*}(1) = A_{CS}^{*} = [p_i(t)|i=1,\cdots,N, t=1,\cdots,T].$

---

## 4.3 Strategy Exploration in the Client-level Sub-game

In the same vein, we model $c_i$'s Client-level sub-game as an MDP, and define the 3-tuple as follows.

- *State:* $s_i(t) \triangleq [acc_i(t)] \in \mathcal{S}_i$, where $acc_i(t)$ is the prediction accuracy of $w_i(t)$ on $D_i$, and $\mathcal{S}_i$ is the state space.
- *Action:* $a_i(t) \triangleq g_i = [\alpha_i(t), \eta_i(t)] \in \mathcal{A}_i$, where $\mathcal{A}_i \equiv \mathcal{G}_i$ is the action space.
- *Reward:* $r_i(t) = R_i(s_i(t), a_i(t), s_i(t+1)) \triangleq u_i(t) = \frac{1}{f_i(\hat{w}_i(t))}$.

Accordingly, $c_i$'s Client-level MDP is defined as:

$$\mathbf{P}_0'\_c_i \ : \ \max_{\mu_i(\cdot)} J_i(\cdot),$$

where $\mu_i(\cdot) : s_i(t) \to a_i(t)$ is the policy, and $J_i(\cdot) = \sum_{t=1}^{T}(\gamma^{t-1} \cdot r_i(t))$.

Similarly, $\mathbf{P}_0'\_c_i$ can be solved by performing Algorithm 2 along with the gaming process, which outputs the appropriate action sequence $A_i^{*} = \left[a_i^{*}(1), \cdots, a_i^{*}(T)\right]$, i.e., the strategy sequence $G_i^{*}(1) = \left[g_i^{*}(1), \cdots, g_i^{*}(T)\right] \equiv A_i^{*}$ in the equilibrium.

---

**Algorithm 2** Local Training Hyper-parameters Tuning

---

**Input:** $\theta_i^{\mu}(\cdot), \theta_i^{Q}(\cdot), \theta_i^{\mu'}(\cdot)$, and $\theta_i^{Q'}(\cdot)$ are $c_i$'s DDPG model parameters; $l_i^{Cri}$ and $l_i^{Act}$ are the learning rates for *Critic* and *Actor* in the MainNet; $\beta_i$ is the tiny updating rate for the TargetNet.

**Output:** $[\alpha_i(t), \eta_i(t)|t=1,\cdots,T]$.

1: **for** $i = 1, \cdots, N$ **do**
2:    Initialize $\theta_i^{\mu}(\cdot), \theta_i^{Q}(\cdot), \theta_i^{\mu'}(\cdot)$, and $\theta_i^{Q'}(\cdot)$;
3:    **for** $t = 1, \cdots, T$ **do**
4:       Observe $s_i(t)$, and hence calculate $r_i(t)$;
5:       Sample $(s_i(\xi), a_{CS}(\xi), r_i(\xi), s_i(\xi+1)), \xi=1,\cdots,|B|$;
6:       **for** $\xi = 1, \cdots, |B|$ **do**
7:          Calculate $y_i(\xi)$ like Line 6, Algorithm 1;
8:       **end for**
9:       Calculate $Loss_{CS}(t-1)$ like Line 8, Algorithm 1;
10:      Update $\theta_i^{Q}(t), \theta_i^{\mu}(t), \theta_i^{\mu'}(t)$, and $\theta_i^{Q'}(t)$ like Line 9, Algorithm 1;
11:   **end for**
12: **end for**
13: **return** $\mu_i^{*}(\cdot)$;
14: **return** $G_i^{*}(1) = A_i^{*} = [\alpha_i(t), \eta_i(t)|t=1,\cdots,T].$

---

## 4.4 Workflow of PAGE

Consequently, we propose PAGE, where $CS$ and $c_i$ collaboratively train global and local models by adaptively adjusting aggregation weights and local training hyper-parameters. To provide an overall insight, we illustrate the $t$-th round of PAGE in Figure 1, and depict the details as follows:

① At the beginning of the $t$-th training round, $CS$ first distributes the global model $W(t)$ to every $c_i$.

② Every $c_i$ initializes the local model $w_i(t)$ as $W(t)$. Then, $c_i$ updates the local DDPG model parameters $\theta_i^{Q}(t), \theta_i^{\mu}(t), \theta_i^{\mu'}(t)$, and $\theta_i^{Q'}(t)$ to generate $\alpha_i(t)$ and $\eta_i(t)$.

③ $c_i$ updates $w_i(t)$ to $\hat{w}_i(t)$ using $\alpha_i(t)$ and $\eta_i(t)$, simply expressed as $\hat{w}_i(t) = Train(\eta_i(t), \alpha_i(t); w_i(t))$.

④ $c_i$ uploads $\hat{w}_i(t)$ to $CS$.

⑤ $CS$ updates the global DDPG model parameters $\theta_{CS}^{Q}(t), \theta_{CS}^{\mu}(t), \theta_{CS}^{\mu'}(t)$, and $\theta_{CS}^{Q'}(t)$ to generate aggregation weights $\{p_i(t)|i = 1, \cdots, N\}$.

⑥ $CS$ aggregates $\hat{w}_i(t), i=1,\cdots,N$ to update the global model as $W(t+1)$ according to Eq. (5).

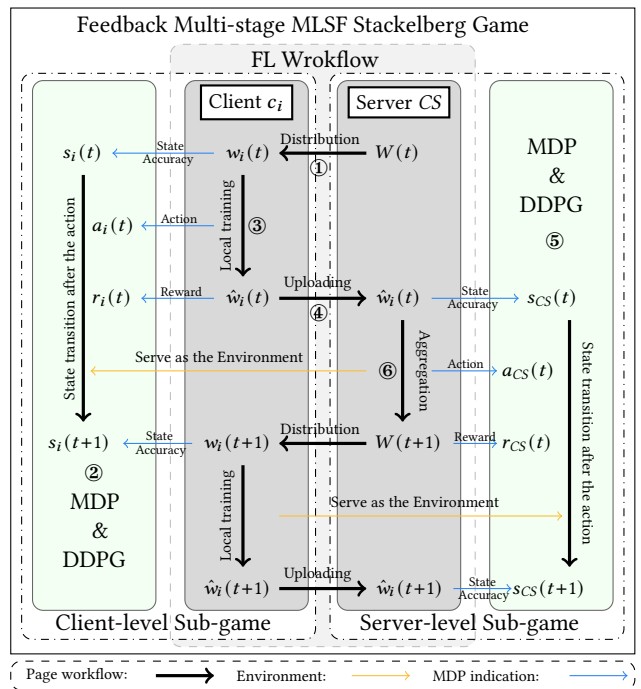

**Figure 1: Workflow of PAGE.**

**PS**: • $CS$ and $c_i$ periodically perform ①-⑥ until $W(t)$ and $w_i(t)$ stop evolving, i.e., achieving the equilibrium of $\mathbf{P}'_0$.
• In the initial training round, the local model training and aggregation are performed by randomly selecting $\alpha_i(t)$, $\eta_i(t)$, and $p_i(t)$, as DDPG models cannot update without prior experience [3].

## 4.5 Theoretical Analysis for the Equilibrium

We then analyze the existence of the equilibrium of $\mathbf{P}'_0$, which is equivalent to the convergence analysis in TFL and PFL.

Before proceeding further, we define the equilibrium of $\mathbf{P}'_0$ in advance. Since the pay-off of every participant in a multi-stage game is an accumulating pursuit, rather than any attained peak, however large, we primarily define the pay-off functions of the sub-games.

**DEFINITION 2 (FOLLOWER'S PAY-OFF FUNCTION [4]).** *The pay-off function of CS is the discounted accumulation of $u_{CS}(t)$ from the $\tau$-th stage, denoted by $U_{CS}(\cdot)$ and defined as:*

$$U_{CS}(G_{CS}(\tau)) = \sum_{t=\tau}^{T} \left(\gamma^t \cdot u_{CS}(t)\right) = \sum_{t=\tau}^{T} \frac{\gamma^t}{F(W(t+1))}, \quad (6)$$

*where $G_{CS}(\tau) = [g_{CS}(\tau), \cdots, g_{CS}(T)], \forall \tau = 1, \cdots, T$ is CS's strategy sequence from the $\tau$-th stage.*

**DEFINITION 3 (LEADER'S PAY-OFF FUNCTION [4]).** *The pay-off function of $c_i$ is the discounted accumulation of $u_i(t)$ from the $\tau$-th stage, denoted by $U_i(\cdot)$ and defined as:*

$$U_i(G_i(\tau)) = \sum_{t=\tau}^{T} \left(\gamma^t \cdot u_i(t)\right) = \sum_{t=\tau}^{T} \frac{\gamma^t}{f_i(w_i(t))}. \quad (7)$$

*where $G_i(\tau) = [g_i(\tau), \cdots, g_i(T)], \forall \tau = 1, \cdots, T$ is $c_i$'s strategy sequence from the $\tau$-th stage.*

Thus, the equilibrium of $\mathbf{P}'_0$ can be defined as follows:

**DEFINITION 4 (FEEDBACK STACKELBERG EQUILIBRIUM (FSE) [4]).** *Given a feedback multi-stage MLSF Stackelberg game $\mathbf{P}'_0$, the feedback stackelberg equilibrium is denoted by $G^*(\tau) = \left[G_1^*(\tau), \cdots, G_N^*(\tau), G_{CS}^*(\tau)\right]$ and defined as:*

$$\begin{aligned} &U_{CS}\left(G^*(\tau)\right) \geq U_{CS}\left(g_{CS}(\epsilon), G^*(\tau)\backslash g_{CS}^*(\epsilon)\right), \\ &U_i\left(G^*(\tau)\right) \geq U_i\left(g_i(\epsilon), G^*(\tau)\backslash g_i^*(\epsilon)\right), \forall i = 1, \cdots, N, \end{aligned} \quad (8)$$

*where $\epsilon$ is the stage index in the range of $[\tau, T]$, $g_i^*(t)$ and $g_{CS}^*(t)$ are optimal strategies for obtaining the maximal utilities at the $t$-th stage, $G_i^*(\tau) = \left[g_i^*(\tau), \cdots, g_i^*(T)\right]$ and $G_{CS}^*(\tau) = \left[g_{CS}^*(\tau), \cdots, g_{CS}^*(T)\right]$ are the optimal strategy sequences from the $\tau$-th stage, and $G^*(\tau)\backslash g_i^*(\epsilon)$ and $G^*(\tau)\backslash g_{CS}^*(\epsilon)$ indicate the optimal strategy sequences except for $g_i^*(\epsilon)$ and $g_{CS}^*(\epsilon)$, respectively.*

Definition 4 expounds that reaching the FSE at which $\mathbf{P}'_0$ ends requires a series of sequential interactions, no matter what stage the measurement starts from.

Based on above definitions, the existence of the FSE of $\mathbf{P}'_0$ can be disclosed by Theorem 1.

**THEOREM 1.** *For $\mathbf{P}'_0$, the feedback stackelberg equilibrium (FSE) $G^*(\tau)$ always exists.*

**PROOF.** We first recall the definition of the value function to measure the strategy in the FSE.

**DEFINITION 5 (VALUE FUNCTION [4]).** *Given $\mathbf{P}'_0$ with the FSE $G^*(\tau)$, let $Z^* = [z^*(\tau), \cdots, z^*(T)]$ be the associated optimal gaming condition trajectory resulting from $z^*(\tau)$. Then, the value functions of CS and $c_i$ are expressed as:*

$$V_{CS}^*\left(z^*(\tau)\right) = \sum_{t=\tau}^{T} \left(\gamma^t \cdot u_{CS}^*(t)\right), \quad (9)$$

*and*

$$V_i^*\left(z^*(\tau)\right) = \sum_{t=\tau}^{T} \left(\gamma^t \cdot u_i^*(t)\right), \forall i = 1, \cdots, N, \quad (10)$$

*where $u_i^*(t)$ and $u_{CS}^*(t)$ are the utilities derived from $g_i^*(t)$ and $g_{CS}^*(t)$, respectively.*

Thus, we can obtain $T - \tau + 1$ sets of value functions. As a result, the only way to confirm the existence of the FSE is to verify whether these value functions satisfy the Bellman equations, shown as:

$$V_{CS}^*(\tau) = \max_{g_1(t), \cdots, g_N(t), g_{CS}(t)} u_{CS}(t) + \gamma \cdot V_{CS}^*\left(z^*(\tau+1)\right), \quad (11)$$

and

$$\begin{aligned} V_i^*(z^*(\tau)) &= \max_{g_1(t), \cdots, g_N(t), g_{CS}(t)} u_i(t) + \gamma \cdot V_i^*\left(z^*(\tau+1)\right), \\ &\forall i = 1, \cdots, N. \end{aligned} \quad (12)$$

Note that the first term on the right side of Eq. (11) highlights the maximal utilities given $Z^*$, and the same to Eq. (12). As a solution, the verification could be achieved through the recursive approach, which is referred to as the *verification theorem* [4]. In other words, the existence of FSE can be confirmed in specific cases for which an explicit solution of the Bellman equations can be obtained, which completes the proof. □

                                        

# 5 EXPERIMENTS AND EVALUATION

## 5.1 Experimental Settings

In this section, we compare PAGE with 10 SOTA baselines, including 5 TFLs, i.e., FedAvg [31], FedProx [25], SCAFFOLD [19], FedDyn [1], and Dap-FL [10], as well as five PFLs, i.e., FEDRECON [33], pFedMe [12], Ditto [24], FedALA [45], and Fed-ROD [9]. The global model generalization and local model personalization are evaluated through the global and local model accuracy over global and local testing sets (defined below), respectively. In particular, the recorded local model accuracy is the average of local model accuracy on clients' corresponding local testing sets. Notably, all presented results are averaged over 3 runs (entire collaborative training processes) with different random seeds.

**Datasets and models:** Our experiments are conducted on four widespread public datasets[2], including Synthetic [8], Cifar-100 [21], Tiny-ImageNet [22], and Shakespeare [31]. For Synthetic, we adopt a multi-class logistic classification model with cross-entropy loss [1]. Also, we adopt ResNet-18 [15] for Cifar-100 and Tiny-ImageNet, and LSTM [16] for Shakespeare. More details of leveraged datasets and corresponding models are summarized in Appendix C.1.

**FL settings and data partition:** By default, our experiments involve 100 clients for the four tasks[3]. For *Logistic on Synthetic*, we use a similar data generation process in [25], where each $c_i$ holds 210 training samples and 90 testing samples on average, and $CS$ holds 7500 testing samples. Clients' samples comprise 30 dimensions of features and 30 classes, and $CS$'s samples cover all features and classes. For *ResNet-18 on Cifar-100* and *ResNet-18 on Tiny-ImageNet*, we divide the original training set into 100 parts uniformly, where the class ratio of each part follows a widely used Dirichlet distribution $Dir(\delta=0.3)$ [36]. Each part is further partitioned as the local training and testing sets on a 7:3 scale, and the original testing/validation set is assigned to $CS$ as the global testing set. For *LSTM on Shakespeare*, we pick the role with more than 8000 sentences as the client, where 4900 and 2100 sentences are used as the local training and testing data, respectively. The remaining sentences of the pricked 100 roles are the global testing data.

**Implementation and Hyperparmeters:** All simulations are implemented on the same computing environment (Linux, 32 Intel(R) Xeon(R) Silver 4108 CPU @ 1.80GHz, NVIDIA GeForce A100, 256GB of RAM and 2T of memory) with Pytorch. In addition, the hyper-parameter settings of PAGE are summarized in Appendix C.2, and baselines are implemented with their original hyper-parameters[4]. We release the codes and datasets at https://github.com/ivy-h7/PAGE.

## 5.2 Results and Evaluation

**Prediction accuracy comparison:** Table 1 illustrates the comparison between PAGE and baselines in terms of global and local model accuracy. As expected, PAGE achieves at most 39.91% gains in terms of local model accuracy, and the global model accuracy

is improved by up to 35.20%. Surprisingly, PAGE comprehensively outperforms all baselines in most cases, where the highest global and local model accuracy is achieved simultaneously, rather than achieving a moderate balance merely. The reason behind this observation is that PAGE integrates the advantages of PFL and TFL methods, to be more specific, local fine-tuning [42] and client selection [29, 37]. Also, we mention that the abnormality concerning global model generalization on Synthetic is attributed to the low degree of *data heterogeneity*, where the global models of baselines could generalize well.

**Communication efficiency comparison:** To explore the communication efficiency of PAGE, we record the convergence round in Table 2. As can be observed, PAGE achieves fewer rounds in most cases, reflecting a more rapid convergence rate and higher communication efficiency. Consequently, PAGE is more competitive in MLaaS, as expensive and rare communication bandwidths are saved in the presence of satisfying the demands of customers and service providers to the greatest extent.

**Origin of performance gains:** In Figure 2, we illustrate the accuracy curves of PAGE together with the reward curves of corresponding DDPG models. One can observe the same variation trends between global/local model accuracy and server/client-side reward curves. It suggests that the server-side DDPG model facilitates global model generalization by adjusting $p_i$ to obtain larger rewards, and client-side DDPG models conduct local training hyperparameter adjustment for expected rewards, benefiting local model personalization. In the same vein, the gains of convergence rates stem from the RL-based adjustment. Besides, global and local models collaboratively evolve into stable conditions, i.e., FSE, which validates the co-opetition intention of PAGE.

**Performance under quantity-skewed heterogeneity:** To test the performance of PAGE facing quantity-skewed data heterogeneity, we conduct unbalanced data partitions on top of the default setting for *ResNet-18 on Cifar-100*, where the ratio of clients' local sample numbers follows logarithmic normal distributions[5] with the mean of 0 and the standard deviation $\sigma=0.1, 0.3$, and $0.5$. In this case, we compare PAGE with PFL in the left part of Table 3. As expected, the global model accuracy are higher than all PFL baselines, while keeping relatively desirable local model personalization. In particular, the global model generalization of PAGE remains stable with the increasing unbalance degree, while PFL becomes worse. Such a property is attributed to the adaptive adjustment of $p_i$.

**Performance under label-skewed heterogeneity:** We then study the effectiveness of PAGE facing label-skewed data heterogeneity for *ResNet-18 on Cifar-100*. The right side of Table 3 illustrates the comparison between PAGE and PFL baselines when adjusting $\delta$ as 0.1, 0.5, and 1 in the default setting. With the label-skewed degree increasing, PFL manifests better local model personalization, but fares less well in global model generalization, which is a somewhat disappointing property in MLaaS. Conversely, PAGE consistently exhibits outstanding personalization, while maintaining generalization. The adjustment of $\eta_i$ and $\alpha_i$ accounts in part for the stable performance.

**Ablation of hyper-parameter tuning completeness:** To understand how the game-based relation contributes to generalization

---

[2]These datasets are collected by the ML community for academic research, and no ethical considerations or legal concerns were violated.

[3]100 is a commonly used client amount to simulate the practical FL implementation in literature. So are 50 and 1000 in the following ablation analysis.

[4]For datasets not involved in original baselines, we provide the appropriate hyper-parameters in our released codes.

[5]A commonly used distribution to calibrate the data quantity [43].

**Table 1: Prediction accuracy comparison between PAGE and baselines. We record the average and variance of global models of 3 runs, as well as the average and variance of clients' local model accuracy. Also, *Improvement* refers to the largest accuracy improvement. Note that *Logistic on Synthetic* cannot be achieved by FedRECON, as the linear layer of the logistic model cannot be partitioned to construct local variables [9].**

| Algorithm | Logistic on Synthetic | | ResNet-18 on Cifar-100 | | ResNet-18 on Tiny-ImageNet | | LSTM on Shakespeare | |
|---|---|---|---|---|---|---|---|---|
| | global acc (%) | local acc (%) | global acc (%) | local acc (%) | global acc (%) | local acc (%) | global acc (%) | local acc (%) |
| More attention on the comparison with local model accuracy of TFL baselines | | | | | | | | |
| FedAvg | 91.46 ±0.07 | 95.26 ±1.26 | 32.97 ±0.03 | 38.30 ±0.44 | 7.85 ±0.04 | 11.29 ±0.74 | 47.52 ±0.07 | 40.24 ±1.45 |
| FedProx | 91.48 ±0.05 | 95.49 ±0.35 | 33.46 ±0.12 | 39.22 ±0.77 | 7.79 ±0.05 | 11.55 ±0.93 | 47.29 ±0.12 | 40.51 ±1.25 |
| SCAFFOLD | 97.37 ±0.08 | 95.71 ±0.53 | 32.81 ±0.03 | 36.12 ±0.81 | 8.39 ±0.02 | 9.17 ±0.94 | 49.14 ±0.06 | 39.36 ±0.49 |
| FedDyn | 97.57 ±0.07 | 94.11 ±1.42 | 33.47 ±0.05 | 35.28 ±1.11 | 7.84 ±0.27 | 11.45 ±1.14 | 51.68 ±0.14 | 42.82 ±0.72 |
| Dap-FL | 92.19 ±0.13 | 94.14 ±1.11 | 32.28 ±0.27 | 40.72 ±1.41 | 8.40 ±0.43 | 11.75 ±2.19 | 51.67 ±0.26 | 48.85 ±1.38 |
| More attention on the comparison with global model accuracy of PFL baselines | | | | | | | | |
| FedRECON | / | / | 24.88 ±0.14 | 31.75 ±0.65 | 6.25 ±0.25 | 10.15 ±1.13 | 38.54 ±0.06 | 35.61 ±2.02 |
| pFedMe | 85.59 ±0.25 | 90.23 ±1.02 | 30.29 ±0.03 | 38.68 ±0.45 | 6.60 ±0.08 | 9.23 ±0.36 | 43.19 ±0.04 | 41.99 ±0.69 |
| Ditto | 92.09 ±0.17 | 95.56 ±1.12 | 31.86 ±0.24 | 39.93 ±1.35 | 7.77 ±0.05 | 9.59 ±0.22 | 48.95 ±0.04 | 47.05 ±0.47 |
| FedALA | 85.51 ±0.04 | 95.42 ±1.07 | 32.10 ±0.05 | 39.63 ±0.84 | 7.63 ±0.05 | 9.83 ±0.66 | 43.45 ±0.09 | 46.77 ±1.14 |
| Fed-ROD | 87.93 ±0.21 | 90.63 ±1.12 | 31.75 ±0.41 | 31.47 ±0.59 | 8.13 ±0.46 | 12.34 ±0.52 | 46.04 ±0.11 | 43.23 ±1.17 |
| **PAGE** | **92.67** ±0.13 | **96.24** ±0.33 | **33.55** ±0.14 | **40.94** ±0.26 | **8.45** ±0.17 | **12.83** ±0.48 | **51.74** ±0.24 | **49.27** ±0.55 |
| *Improvement* | 8.37 | 6.66 | 34.85 | 30.09 | **35.20** | **39.91** | 34.25 | 38.36 |

**Table 2: Convergence round of PAGE and baselines. Convergence round refers to the round that the global (averaging local) model accuracy stops increasing for TFL (PFL). The column of PAGE records the round at which the FSE achieves.**

| Task | **PAGE** | FedAvg | FedProx | SCAFFOLD | FedDyn | Dap-FL | FedRECON | pFedMe | Ditto | FedALA | Fed-ROD |
|---|---|---|---|---|---|---|---|---|---|---|---|
| Synthetic | 891 | 902 | 896 | 878 | 901 | 900 | / | 843 | 491 | 501 | 497 |
| Cifar-100 | 499 | 510 | 497 | 540 | 502 | 337 | 641 | 550 | 313 | 506 | 401 |
| Tiny-ImageNet | 404 | 430 | 479 | 422 | 366 | 402 | 361 | 513 | 490 | 523 | 493 |
| Shakespeare | 602 | 552 | 655 | 546 | 607 | 590 | 657 | 642 | 498 | 646 | 556 |

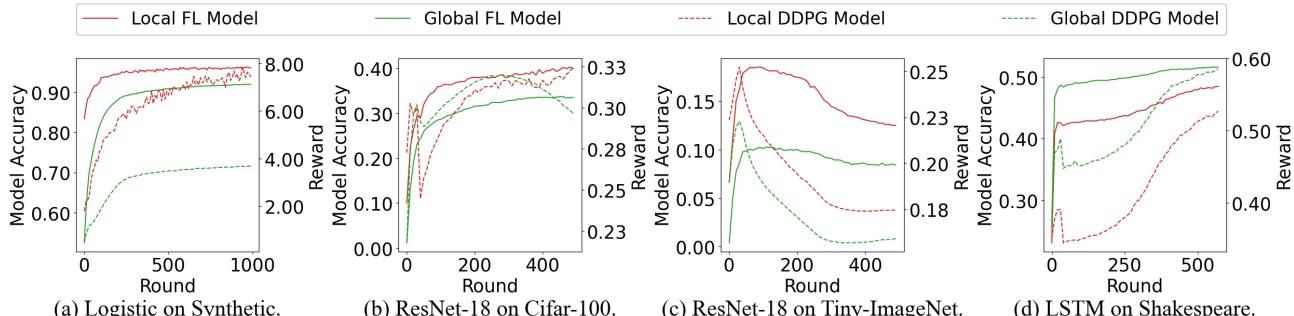

(a) Logistic on Synthetic.  (b) ResNet-18 on Cifar-100.  (c) ResNet-18 on Tiny-ImageNet.  (d) LSTM on Shakespeare.

**Figure 2: Model accuracy curves of PAGE together with corresponding DDPG reward curves. The left y-axes calibrate the model accuracy of FL models (solid curves), and the right y-axes calibrate the rewards of DDPG models (dotted curves).**

and personalization, we conduct ablation analyses for *ResNet-18 on Cifar-100* by adjusting one or two factors in PAGE, while other factors remain constant. In Figure 3, only adjusting $p_i$ benefits the global model performance, while adjusting $\eta_i$ or $\alpha_i$ promotes the local model performance. By contrast, simultaneously adjusting $\eta_i$ and $\alpha_i$ achieves higher local model accuracy and more rapid convergence rates than solely adjusting one factor. In addition, compared to the equilibrium in the setting of remaining $\eta_i$ or $\alpha_i$ constant, PAGE's equilibrium has better generalization and personalization. Thus, the completeness of balance-controlling factors is confirmed.

**Ablation of client amount:** The top part of Table 4 explores the performance with different client amounts for *Logistic on Synthetic*. Seemingly, the balance between global and local models would not change with the client amount increasing, but requires more rounds. But we mention that the increasing round with the increasing client amount widely exists in diverse FL methods rather than merely in PAGE. The reason behind this attribute is that more participants would expand the feature space of local data, which exacerbates the difficulty of achieving the equilibrium (convergence in TFL/PFL).

**Table 3: Comparison between PAGE and PFL under different data heterogeneity. Smaller $\sigma$ reflects lower unbalance data distributions, and smaller $\delta$ indicates heavier label skew.**

| Algorithm | Quantity Skew – acc (%) | | | | | | Label Skew – acc (%) | | | | | |
|---|---|---|---|---|---|---|---|---|---|---|---|---|
| | $\sigma = 0.1$ | | $\sigma = 0.3$ | | $\sigma = 0.5$ | | $\delta = 0.1$ | | $\delta = 0.5$ | | $\delta = 1$ | |
| | global | local | global | local | global | local | global | local | global | local | global | local |
| **PAGE** | **33.47** | **40.96** | **33.61** | **40.91** | **33.52** | **40.93** | **32.69** | **54.58** | **33.57** | **40.97** | **33.53** | **40.02** |
| FedRECON | 24.13 | 31.88 | 22.91 | 32.91 | 21.23 | 34.95 | 23.77 | 47.29 | 25.24 | 25.65 | 25.69 | 20.13 |
| pFedMe | 30.22 | 38.92 | 30.18 | 39.45 | 30.09 | 39.68 | 28.26 | 47.23 | 31.36 | 33.28 | 31.47 | 29.62 |
| Ditto | 31.46 | 39.93 | 30.96 | 39.93 | 30.31 | 39.95 | 30.59 | 53.50 | 32.47 | 36.46 | 32.79 | 33.24 |
| FedALA | 31.55 | 39.63 | 31.46 | 39.64 | 31.08 | 39.64 | 30.46 | 53.58 | 32.27 | 34.98 | 32.84 | 30.24 |
| Fed-ROD | 31.71 | 32.19 | 31.39 | 32.31 | 31.06 | 32.88 | 30.53 | 49.44 | 32.35 | 28.18 | 32.16 | 22.98 |

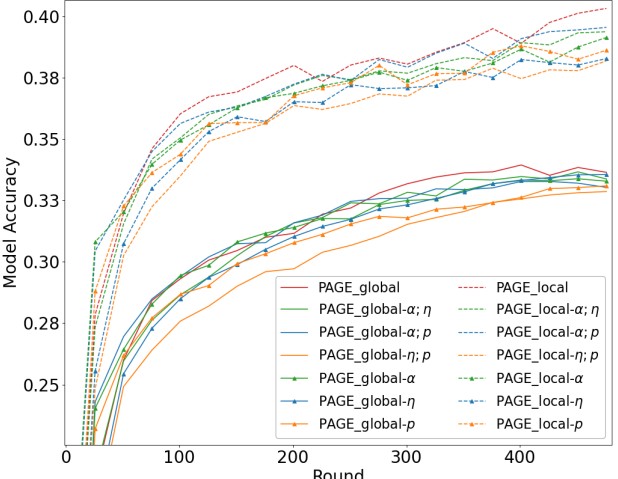

**Figure 3: Completeness of balance-controlling factors.**

This provides an instructive insight into product FL with enormous clients, i.e., the practical implementation of PAGE at scale.

**Table 4: Exploration of other properties. The last column refers to the round achieving equilibrium.** 50 and 1000 indicate default settings with distinct client amounts, and 10 : 1 and 1 : 10 are the ratios between global and local rewards.

| Task | global acc (%) | local acc (%) | Round |
|---|---|---|---|
| Client amount (Synthetic) | | | |
| PAGE-50 | 91.75 | 96.62 | 618 |
| PAGE-1000 | 92.39 | 96.43 | 928 |
| **PAGE (100)** | 92.67 | 96.24 | 891 |
| Generalization or personalization trend (Cifar-100) | | | |
| PAGE-10:1 | 35.11 | 40.19 | 538 |
| PAGE-1:10 | 31.91 | 41.55 | 493 |
| **PAGE** | 33.55 | 40.94 | 499 |

**Bias between generalization and personalization:** Also, we discuss the biased variant of PAGE for *ResNet-18 on Cifar-100*, where the reward ratio[6] between the server-side DDPG and the client-side

---

[6]The ratios are empirical settings in our simulation, which, for reproducibility, could be adjusted with the changes in the bias degree, client amount, task, etc.

DDPG varies to simulate the varying biases between generalization and personalization in practice. As shown in the bottom part of Table 4, PAGE could tip the balance to an expected side by changing the reward ratio according to the market demand in MLaaS. Particularly, by comparing the results with baselines in Table 1, the biased variants of PAGE outperform all TFL/PFL baselines in terms of corresponding global/local model accuracy and convergence rates.

**Computation efficiency:** Besides, we record the computation performance of the main operations of PAGE in Table 5, where the DDPG model training efficiency is higher than FL models by an order of magnitude. Also, the model size of the DDPG model is significantly smaller than FL models in practice, such as prevailing large language models. It suggests that PAGE is efficient in terms of computation, as the DDPG model training could be accomplished rapidly during the entire collaborative training process. Besides, DDPG can be implemented on CPU rather than rarer GPU resources, which highlights the technical feasibility of PAGE.

**Table 5: Computation performance of main operations.**

| Index | Operation | Time (ms/Byte) |
|---|---|---|
| 1 | Local training | $1.45 \times 10^{-4}$ |
| 2 | Model aggregation | $1.93 \times 10^{-6}$ |
| 3 | Local DDPG training | $6.84 \times 10^{-5}$ |
| 4 | Global DDPG training | $7.38 \times 10^{-5}$ |

## 6 CONCLUSION AND FUTURE WORK

PAGE is the first FL algorithm that balances the local model personalization and global model generalization. A key insight into developing PAGE is that an iterative co-opetition exists between the server and clients, which runs parallel with a feedback multi-stage MLSF Stackelberg game. Particularly, the server/client-level sub-games and MDPs have uncanny resemblances. As such, PAGE introduces DDPG to solve the equilibrium of the formulated game, thereby providing a stable terminating condition for FL, i.e., the balance between personalization and generalization.

As a future work, we will take the security and privacy issues into account. In addition, by jointly considering resource heterogeneity, a variant of PAGE could be implemented in a more practical scenario, which is already investigated in Appendix D theoretically. We leave the empirical validation in the future.

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

## A APPENDIX: RELATED WORKS

Since the main battlefield against *data heterogeneity* is PFL, our literature review mainly focuses on its two major categories: data-based and model-based methods. In general, data-based methods emphasize reducing statistical divergence among local datasets, while model-based methods aim to tailor customized models for diverse clients.

To be specific, data augmentation [13, 23] is a common data-based method, which generates proxy datasets based on shared local data distribution for model training, reducing client drift. However, sharing local data distribution is viewed as a violation of privacy and data minimization [36]. Another line of data-based methods is client selection [29, 37], where selected datasets approach homogeneous data distribution to heighten model personalization. Although client selection is equivalent to the Server-level sub-game if adjusting some aggregation weights to zero, it never occurs in PAGE, thereby ensuring extensive local data feature capture and system fairness.

By contrast, model-based PFL methods are more practical, which mainly consist of local fine-tuning [39, 42], transfer learning [44], meta-learning [18, 33], regularization [12, 14, 24], knowledge distillation [11, 46], etc. The dominant method is local fine-tuning [39, 42], where clients fine-tune the global model through several gradient descent steps over their local datasets. A variant of local fine-tuning is transfer learning [44], which takes advantage of the knowledge extracted by the global model to employ personalized models for different clients. Similarly, meta-learning methods [18, 33] could be regarded as another variant of local fine-tuning, where local adaptation is achieved by mapping the meta-testing step to the personalization process. However, the main drawback of local fine-tuning (including transfer learning and meta-learning) is that the personalized model is pruned to overfitting, whereas the other important part, global model generalization, is deteriorated. Although the Client-level sub-game of PAGE is parallel to local fine-tuning to some extent, global model generalization is taken into account in the Server-level sub-game, which mitigates the overfitting drawback. Regularization methods [12, 14, 24] introduce new formulations for PFL, where regularizers on the distance of local and global models are added to control the optimization degree. Also, knowledge distillation methods [11, 46] apply regularizers on the predictions between local and global models in a teacher-student paradigm. Yet, regularizers have been proven less effective by Chen *et al.* [9]. Besides, multi-task learning [30], model mixture [14, 24], clustering [11, 30, 41], and model decoupling [17, 35] could be leveraged to achieve PFL as well. For more details, we recommend Tan *et al.*'s survey [36].

## B APPENDIX: NOTATION SUMMARY

For clarity, we summarize important notations in Table 6.

## C APPENDIX: EXPERIMENTS & EVALUATION

### C.1 Details of leveraged datasets and models

**Synthetic** [8] is a classification dataset, where two factors control how much the local data at each client differs from that of others. In our experiments, we generate 210 training samples and 90 testing samples for each client, as well as 7500 testing samples for the

**Table 6: Notations in PAGE**

| Notations | Definition |
|---|---|
| $c_i, i = 1, \cdots, N$ | $N$ clients; Leaders in the game |
| $CS$ | Central server; Follower in the game |
| $t = 1, \cdots, T$ | Round; Stage in the game |
| $D_i$ | Private local dataset |
| $D_{CS}$ | Public global dataset |
| $f_i(\cdot)$ | Local loss function |
| $F(\cdot)$ | Global loss function |
| $w_i, w_i^*$ | Local model and optimal local model |
| $W, W^*$ | Global model and optimal global model |
| $p_i$ | Aggregation weight |
| $\alpha_i$ | Local training epoch |
| $\eta_i$ | Local learning rate |
| $g_i, g_{CS}$ | Strategies of $c_i$ and $CS$ |
| $u_i, u_{CS}$ | Utilities of $c_i$ and $CS$ |
| $s_i, s_{CS}$ | States of $c_i$ and $CS$ |
| $a_i, a_{CS}$ | Actions of $c_i$ and $CS$ |
| $r_i, r_{CS}$ | Rewards of $c_i$ and $CS$ |
| $\mu_i(\cdot), \mu_{CS}(\cdot)$ | Policies of $c_i$ and $CS$ |
| $\gamma$ | Discount factor |
| $\theta_i^Q(t), \theta_i^\mu(t), \theta_i^{\mu'}(t), \theta_i^{Q'}(t),$ $\theta_{CS}^Q(t), \theta_{CS}^\mu(t), \theta_{CS}^{\mu'}(t), \theta_{CS}^{Q'}(t)$ | DDPG model parameters of $c_i$ and $CS$ |

central server. Clients' samples comprise 30 dimensions of features and 30 classes, and the samples held by the server cover all features and classes.

**Cifar-100** [21] is an image classification dataset consisting of $30\times30\times3$ color images in 100 classes, with 600 images per class. Each class has 500 training samples and 100 testing samples.

**Tiny-ImageNet** [22] is an image classification dataset containing 100000 samples of 200 classes (500 for each class) downsized to $64\times64\times3$ color images. Each class has 500 training images, 50 validation images, and 50 test images.

**Shakespeare** [31] is built from *The Complete Works of William Shakespeare* by treating each role in the play as a client, and their dialogue lines as the samples. The main task of this dataset is for next-character prediction.

**Multi-class classification model** [1] has a linear layer, which takes each 30-dimensional input sample of Synthetic as the input, and outputs a 30-dimensional vector. In addition, the cross-entropy loss is used as the loss function.

**LSTM** [16, 25] consists of an 8-dimensional embedding layer, two LSTM layers with a hidden size of 100 units, and a fully connected output layer. For each input sequence sample, the model embeds 80 characters into a learned 8-dimensional space, and outputs one character.

**ResNet-18** [15] consists of a $7 \times 7$ convolutional layer, two pooling layers, eight residual units, and one fully connected layer. The loss function is set as the cross-entropy loss. In particular, the Batch Normalization after the convolutional operation is substituted by Group Normalization. For each input sample, ResNet-18 outputs a 100-dimensional vector for Cifar-100 and a 200-dimensional vector for Tiny-ImageNet, respectively.

## C.2 Default hyper-parameter settings

For reproducibility, we summarize the default hyper-parameter settings of PAGE across the four tasks in Table 7.

**Table 7: Default hyper-parameter settings of PAGE.**

| Task | Synthetic | Cifar-100 | Tiny-ImageNet | Shakespeare |
|------|-----------|-----------|---------------|-------------|
| $l_{CS}^{Cri}$ | 0.01 | 0.05 | 0.001 | 0.005 |
| $l_{CS}^{Act}$ | 0.001 | 0.005 | 0.0001 | 0.0005 |
| $\beta_{CS}$ | 0.01 | 0.05 | 0.001 | 0.005 |
| $l_i^{Cri}$ | 0.01 | 0.05 | 0.001 | 0.005 |
| $l_i^{Act}$ | 0.001 | 0.005 | 0.0001 | 0.0005 |
| $\beta_i$ | 0.01 | 0.05 | 0.001 | 0.005 |

## D APPENDIX: RESOURCE HETEROGENEITY

### D.1 Time-varying resource consumption model

Following [10], our resource consumption model considers clients' *time-varying* resource consumption but neglects $CS$, as $CS$ usually has sufficient resources for stable aggregation and communication. In particular, the resource consumption comprises computation and communication consumption:

• *Computation resource consumption.* In FL, $c_i$'s local training process only occurs on the computation unit of its local device. Assuming the frequency of $c_i$'s computation unit in the $t$-th training round is $\Gamma_i(t)$, $c_i$'s computation resource consumption for one epoch of local training is denoted by $E_i^{cmp}(t)$ and defined as:

$$E_i^{cmp}(t) = |D_i| \cdot \kappa_i \cdot \Lambda_i(t) \cdot \Gamma_i^2(t), \tag{13}$$

where $\Lambda_i(t)$ is the number of computation unit cycles per sample, and $\kappa_i$ is the effective switched capacitance.

• *Communication resource consumption.* The communication of FL hinges on the general wireless communication system, e.g., the orthogonal frequency division multiple access (OFDMA). In detail, the communication of $c_i$ refers to uploading local models and downloading the global model. Thus, in the light of the Shannon formula, the uplink and downlink communication resource consumption could be denoted by $E_i^{com}(t)$ and $E_i'^{com}(t)$, and defined as:

$$E_i^{com}(t) = \frac{\rho_i(t) \cdot \Upsilon}{\Psi_i(t) \cdot \log_2\left(1 + \frac{\phi_i(t)\rho_i(t)}{\Delta_i(t)\Psi_i(t)}\right)}, \tag{14}$$

and

$$E_i'^{com}(t) = \frac{\rho_i'(t) \cdot \Upsilon}{\Psi_i'(t) \cdot \log_2\left(1 + \frac{\phi_i'(t)\rho_i'(t)}{\Delta_i'(t)\Psi_i'(t)}\right)}, \tag{15}$$

where $\rho_i(t)$ and $\rho_i'(t)$ are the transmission power, $\Psi_i(t)$ and $\Psi_i'(t)$ are the allocated communication bandwidths, $\phi_i(t)$ and $\phi_i'(t)$ are the channel gains, $\Delta_i(t)$ and $\Delta_i'(t)$ are the power spectral density of noise for uplink and downlink, respectively, and the local model and the global model share the same size $\Upsilon$. Note that the consumption of uplink is much larger than downlink in practice.

A ground truth is that $c_i$'s total resource for computation and communication in any given round is limited, expressed as a resource constraint:

$$\alpha_i(t) \cdot E_i^{cmp}(t) + E_i^{com}(t) + E_i'^{com}(t) \leq E_i(t), \tag{16}$$

where $E_i(t)$ is the upper bound of the total resource in the $t$-th training round. Thus, by jointly considering $\mathbf{P_0}$ and resource constraints during the whole FL procedure, we further formulate FL as an optimization problem with constraint conditions, defined as:

$$\mathbf{P_1} : \begin{cases} W^* = \underset{W}{\arg\min} \left\{ F(W) := \sum_{i=1}^{N} (p_i \cdot f_i(w_i)) \right\}, \\ w_i^* = \underset{w_i}{\arg\min} \{f_i(w_i)\}, i = 1, \cdots, N, \end{cases}$$

$$\text{s.t. } \alpha_i(t) \cdot E_i^{cmp}(t) + E_i^{com}(t) + E_i'^{com}(t) \leq E_i(t),$$

$$t = 1, \cdots, T.$$

Note that $E_i^{cmp}(t)$, $E_i^{com}(t)$, and $E_i'^{com}(t)$ vary with time, as they are strongly impacted by practical time-varying factors, such as temperature, battery power, etc.

### D.2 Strategy exploration under resource heterogeneity

In practice, heterogeneous local resources restrict the model training process to a great extent, thereby significantly impacting the balance that PAGE achieves. Particularly, clients with limited computation or communication resources might fail to complete the local training or model uploading using the obtained optimal strategy within a stipulated time window. Fortunately, *resource heterogeneity* only restricts the operations of clients instead of $CS$, as $CS$ usually has ample computation and communication resources. Thus, if formulating $\mathbf{P_1}$ as a feedback multi-stage MLSF Stackelberg game, the Server-level sub-game $\mathbf{P_1'}\_CS$ and its strategy exploration process would be totally the same as $\mathbf{P_0'}\_CS$. Yet, the Client-level sub-game of every $c_i$ should incorporate the resource constraints, denoted by $\mathbf{P_1'}\_c_i$ and defined as:

$$\mathbf{P_1'}\_c_i : \max_{\mu_i(\cdot)} J_i(\cdot),$$

$$\text{s.t. } b_i(t) \leq 0, \ t = 1, \cdots, T,$$

where $b_i(t) = \alpha_i(t) \cdot E_i^{cmp}(t) + E_i^{com}(t) + E_i'^{com}(t) - E_i(t)$.

In contrast to $\mathbf{P_0'}\_c_i$, $\mathbf{P_1'}\_c_i$ cannot be solved by DDPG directly, as the complexity is significantly exacerbated in domains where $J_i(\cdot)$ and the constraints require joint optimization. Therefore, we convert $\mathbf{P_1'}\_c_i$ into its Lagrangian dual problem [2] $\tilde{\mathbf{P}}_1'\_c_i$, expressed as:

$$\tilde{\mathbf{P}}_1'\_c_i : \min_{\lambda_i(t)} \max_{\mu_i(\cdot)} \tilde{J}_i(\cdot), \ t = 1, \cdots, T,$$

where $\tilde{J}_i(\cdot) = J_i(\cdot) - \sum_{t=1}^{T} (\lambda_i(t) \cdot b_i(t))$, and $\lambda_i(t) \geq 0$ is the Lagrangian multiplier.

Hence, $\tilde{\mathbf{P}}_1'\_c_i$ could be optimized by DDPG with some modifications. In detail, on the one hand, $J_i(\cdot)$ is changed to $\tilde{J}_i(\cdot)$ when updating $\theta_i^\mu(t)$. On the other hand, after updating the DDPG model parameters in every $t$-th training round, $\lambda_i(t)$ should be updated accordingly, shown as:

$$\lambda_i(t) = \lambda_i(t-1) - l_i^{Lag} \cdot \nabla_{\lambda_i} \tilde{J}_i(t-1), \tag{17}$$

where $l_i^{Lag}$ is the learning rate, and $\nabla_{\lambda_i} \tilde{J}_i(t-1)$ is the gradient.

## D.3 Theoretical analysis of the conversion

The rationality of the conversion from $\mathbf{P}'_1\_c_i$ to $\tilde{\mathbf{P}}'_1\_c_i$ mainly corresponds to the existence of the Lagrangian dual problem and the equivalence of the two problems. For the existence, we first provide some basic definitions and assumptions.

**DEFINITION 6.** *The lowest-consumption policy, which generates actions consuming minimum resources in each training round, is denoted by $\mu_i^{low}(\cdot)$ and defined as:*

$$\mu_i^{low}(\cdot) = \arg\min_{\mu_i(\cdot)} H(a_i^{low}(t)), t = 1, \cdots, T, \qquad (18)$$

*where $a_i^{low}(t)$ is the action consuming minimum resources, $H(a_i^{low}(t)) = \alpha_i^{low}(t) \cdot E_i^{cmp}(t) + E_i^{com}(t) + E_i'^{com}(t)$ is the resource consumed by $a_i^{low}(t)$, and $\alpha_i^{low}(t)$ is the local training epoch in $a_i^{low}(t)$.*

**DEFINITION 7.** *The highest-consumption policy, which generates actions consuming maximal resources in every training round, is denoted by $\mu_i^{high}(\cdot)$ and defined as:*

$$\mu_i^{high}(\cdot) = \arg\min_{\mu_i(\cdot)} H(a_i^{high}(t)), t = 1, \cdots, T, \qquad (19)$$

*where $a_i^{high}(t)$ is the action consuming maximal resources, $H(a_i^{high}(t)) = \alpha_i^{high}(t) \cdot E_i^{cmp}(t) + E_i^{com}(t) + E_i'^{com}(t)$ is the resource consumed by $a_i^{high}(t)$, and $\alpha_i^{high}(t)$ is the local training epoch in $a_i^{high}(t)$.*

**ASSUMPTION 1.** *The total resources under the lowest-consumption policy are strictly feasible, expressed as:*

$$H(a_i^{low}(t)) < E_i(t), t = 1, \cdots, T. \qquad (20)$$

**ASSUMPTION 2.** *The consumed resources exceed the total resources under the highest-consumption policy, expressed as:*

$$H(a_i^{high}(t)) > E_i(t), t = 1, \cdots, T. \qquad (21)$$

Thus, the existence of the Lagrangian dual problem can be drawn in Theorem 2.

**THEOREM 2.** *The optimal value of $\lambda_i(t)$ always exists and is positive under Assumption 1 and Assumption 2.*

**PROOF.** In practice, Assumption 1 and Assumption 2 are naturally held. On the one hand, Assumption 1 discloses that any potential policy of $\tilde{\mathbf{P}}'_1\_c_i$ would not derive an over-exceeded resource consumption. Therefore, $\mathbf{P}'_1\_c_i$ and $\tilde{\mathbf{P}}'_1\_c_i$ always have a strictly feasible solution. On the other hand, under Assumption 2, a positive optimal Lagrangian multiplier would penalize the violation of the total resource constraint. Conversely, if the consumption of $\mu_i^{high}(\cdot)$ does not exceed $E_i(t)$, $\tilde{\mathbf{P}}'_1\_c_i$ degrades into $\mathbf{P}'_0\_c_i$, which completes the proof. □

Theorem 2 not only discloses the existence of the Lagrangian dual problem of $\mathbf{P}'_1\_c_i$, but also provides a prerequisite to the equivalence of the Lagrange-based conversion. Consequently, the equivalence of such a Lagrange-based conversion from $\mathbf{P}'_0\_c_i$ to $\tilde{\mathbf{P}}'_1\_c_i$ is given in Theorem 3.

**THEOREM 3.** *For $\mathbf{P}'_1\_c_i$ and $\tilde{\mathbf{P}}'_1\_c_i$ with any $\lambda_i(t)$, there exists a feasible $\mu_i(\cdot)$, such that $J_i^*(\cdot) \le \tilde{J}_i^*(\cdot)$, where $J_i^*(\cdot)$ and $\tilde{J}_i^*(\cdot)$ are the optimal values of $J_i(\cdot)$ and $\tilde{J}_i(\cdot)$, respectively.*

**PROOF.** We first recall some basic definitions and lemmas.

**DEFINITION 8 (SLATER CONDITION).** *[34] For $\mathrm{P}'_1\_c_i$ or $\tilde{\mathrm{P}}'_1\_c_i$, there exists a $\mu_i(\cdot) \in \mathrm{relint}\left(\bigcap_{t=1}^{T} dom\left(b_i(t)\right)\right)$, such that $b_i(t) < 0, t = 1, \cdots, T$.*

**LEMMA 1 (STRONG DUALITY).** *[7] Suppose that Slater condition holds, and $J_i(\cdot)$ is convex, $\tilde{J}_i^*(\cdot) = J_i^*(\cdot)$.*

**LEMMA 2 (WEAK DUALITY).** *[7] $J_i^*(\cdot)$ is upper bounded by $\tilde{J}_i^*(\cdot)$, i.e., $J_i^*(\cdot) \le \tilde{J}_i^*(\cdot)$.*

Since $J_i(\cdot)$ is defined as the discounted accumulation of immediate rewards based on the loss function of FL, the convexity of $J_i(\cdot)$ is ambiguous. Thus, we discuss the equivalence of $\mathbf{P}'_1\_c_i$ and $\tilde{\mathbf{P}}'_1\_c_i$ by category.

If $J_i(\cdot)$ is convex, according to Lemma 1, $\tilde{\mathbf{P}}'_1\_c_i$ is apparently equivalent to $\mathbf{P}'_1\_c_i$. In contrast, if $J_i(\cdot)$ is non-convex, the *Strong Duality* does not hold anymore. Still, according to Lemma 2, the convexity can be loosed. Therefore, the optimal value of $\tilde{\mathbf{P}}'_1\_c_i$ is approximate to the optimal value of $\mathbf{P}'_1\_c_i$, which completes the proof. □

