# OpenReview forum: "PAGE: Equilibrate Personalization and Generalization in Federated Learning"
_ACM.org/TheWebConf/2024/Conference — TheWebConf24_

### Official Review · Reviewer_YneF · 2023-11-25

**Novelty:** 4
**Technical Quality:** 5

**Review:**

Federated learning (FL) enables numerous clients to jointly train a central machine learning model with the support of a central server. Nonetheless, existing FL techniques address the challenges of handling data disparities among clients by treating local model customization and global model generalization as distinct concerns. In this research paper, the authors have introduced a novel game theory-based strategy to achieve a harmonious blend of personalization and generalization.

Strengths:
1) Thorough experiments illustrate the effectiveness of the proposed method.

Weaknesses:
1) The paper is hard to follow.
2) We have no information regarding the overhead of current methods.

**Questions:**

1) The authors in the paper use a lot of space to show various equations, without providing enough information to provide the motivation/insights of these equations. As a result, it is hard for the readers to follow the paper.
2) Table 5 only shows the computation cost of the proposed method. Please also show the overhead of current methods for better comparison.

**Reviewer Confidence:**

3: The reviewer is confident but not certain that the evaluation is correct

**Scope:**

2: The connection to the Web is incidental, e.g., use of Web data or API

---

### Official Review · Reviewer_KdRG · 2023-11-27

**Novelty:** 6
**Technical Quality:** 5

**Review:**

This paper explores the problem of how to balance personalization and globalization of a federated learning model. It introduces a game-theoretic algorithm called PAGE, that re-envisions the FL process as a cooperative competition, Feedback Multi-stage MLSF Stackelberg game between the participating FL clients and the server. The PAGE algorithm finds equilibrium between local model personalization and global model generalization using Markov decision processes, and by leveraging a reinforcement learning algorithm.

The paper proposes a novel approach to improving the final model produced by a standard FL process, by taking a step back and re-thinking the overall goal of the process followed, and that past papers have primarily focused on either producing an optimal global model, or an optimal local model (per client), but never tried to find a good balance between the two.

The process is theoretically explained in depth, and experimentally tested on various datasets used in ML and FL literature.

What i am missing from this paper, and given the complex game-theoretic approach proposed, is a high level overview of the method which would help the reader understand the basic process followed, which agent in the game does what, and why. I only find with some steps in section 4.4 which help with this, but it is not enough and does not cover the full method and how it is incorporated in the overall FL process.

A few remarks about the experiments:
- There seems to be a strange performance of the Res-Net-18 on Tiny-ImageNet. Why is the performance dropping after ~200 FL rounds?
- It would be best to introduce B/W-friendly figures for the experiments (at least).
- Figure 3 needs better explanation: what specific values (or range of values) do the parameters take? Are the model accuracy  (y-axis) for local model performance an average of all local models? How are they calculated? By executing the test accuracy of local models on the server test set holdout?

**Questions:**

See questions / remarks in review section.

**Reviewer Confidence:**

3: The reviewer is confident but not certain that the evaluation is correct

**Scope:**

4: The work is relevant to the Web and to the track, and is of broad interest to the community

---

### Official Review · Reviewer_Chht · 2023-11-30

**Novelty:** 4
**Technical Quality:** 4

**Review:**

The authors address a new issue in federated learning, namely the balance between personalization and generalization. The PAGE algorithm based on game theory is introduced, the game is expressed as a Markov decision-making process, and the reinforcement learning algorithm is used to solve it. The structure of the paper is well organized and makes the argument easy to understand. Their extensive experiments on multiple datasets show improved overall performance. However, the decision regarding the balance between personalization and generalization may require further exploration and explanation. A fairer and more comprehensive comparison with more work on related or similar problems can be further studied. The proposed algorithm is not very clear in terms of details and there are still some doubts, and the authors should provide further explanation and clarification. Besides, I recommend that the authors conduct further research into the impact of the inherent volatility of reinforcement learning methods in order to more convincingly demonstrate the applicability of the proposed method in the real world.

Pros:
	New and significant issue addressed: The paper tackles the crucial issue of balancing personalization and generalization in federated learning.
	Novel approach: The introduction of the PAGE algorithm, based on game theory and reinforcement learning, is an innovative way to solve this problem.
	High clarity: The paper is well-organized and the argument is easy to understand.
	Extensive experiments: Comprehensive testing on several datasets show an improved performance.

Cons:
	The author's strategy design in the action space violates the basic principle of global loss F(·)  in federated learning. Specifically, the action space of the server determines the weight of each client model aggregation, and this weight is dynamic following the training process. The reward function used to evaluate the utility of states and actions relies on the Public global dataset D_cs. This leads to the fact that the loss F(·) of the final aggregated global model does not actually reflect the global data distribution, but simply trains multiple models, in which the local personalized model is suitable for local data, while the global model is only used to measure the Public global dataset D_cs. Moreover, the performance of the finally trained model strongly depends on the setting of the Public global dataset D_cs.
	When customizing the action space of the client, each client separately determines the learning rate μ and epoch of the local update, which results in the local update of each client possibly being asynchronous. In the action space, the limits on the range of these parameters are not clear. Also, it is not clear whether the asynchronicity of updates leads to the drift of the global gradient.
	The decision regarding the balance between personalization and generalization could be further explored and explained. The authors mention in the introduction that“To extend their insight, we specify that personalization and generalization share equal status in FL, and the balance between them is much needed.” More evidence or experiments need to be added to prove that equal status is better. Because making a trade-off between the two to achieve optimal results is not the same as giving the same weight and status to both. Moreover, there actually are many similar research works that modify the training process, such as adding regularization terms or masks, to obtain a global model with stronger generalization and a local personalized model at the same time. They are more technically convincing and may look better than this paper. Authors may consider comparing with these works:
	Federated Continual Learning with Weighted Inter-client Transfer
	Domain-Adversarial Training of Neural Networks
	Reinforcement learning requires sufficient exploration and updating to obtain an effective policy network. In the author's experiments and algorithms, as described in step 5 of PAGE, the DDPG policy network of the central server and local clients is trained and updated in federated learning. Can he really achieve better decision-making effects in the early stages of training? The use of reinforcement learning introduces an element of volatility, creating uncertainty about the method's real-world applicability.
	Are the results in Table 1 for fixed rounds or after all models have converged? The result of figure 2(c) shows that the model training seems to be overfitting and has not converged to the optimal result. It is not fair if the authors only compare the model accuracy of different methods within a fixed round of their own choosing. There are many methods that can quickly increase model accuracy in a short period of time, but the final performance is not necessarily good.

**Questions:**

Same to the Cons, and also the following additional comments:
	It is necessary to explain in detail how to deal with the gradient drift caused by the asynchrony of updates, and how to set the range of the action space in the experiment.
	How was the Public global dataset D_cs set in the experiment? I hope the author will add an experiment to study the relationship between the scale of D_cs and model performance.
	I hope the authors can add more comparisons of related work or methods that can be used to deal with similar problems in the introduction and experiments.
	The authors may add more indicators of variance to the experimental results, not just in Table 1.

**Reviewer Confidence:**

3: The reviewer is confident but not certain that the evaluation is correct

**Scope:**

3: The work is somewhat relevant to the Web and to the track, and is of narrow interest to a sub-community

---

### Official Review · Reviewer_nXUN · 2023-11-30

**Novelty:** 5
**Technical Quality:** 4

**Review:**

1- Summary
The paper introduces PAGE, an innovative algorithm designed to balance personalisation and generalisation in Federated Learning (FL). It addresses the critical challenge of data heterogeneity by formulating the problem as a co-opetition game between clients and a server, using game theory and Markov decision processes. This approach simplifies the complexity of finding an equilibrium, and the experiments demonstrate significant improvements in both global and local prediction accuracy across various datasets.

2- Presentation
2.1 Clarity and Structure
The paper is well-structured with clear sections, including Abstract, Introduction, Related Work, Problem Statement, and Methodology. However, it could benefit from improved clarity in the explanation of complex concepts, especially in the methodology and algorithm sections.

2.2 Readability
The paper is generally very readable and well presented, following a logical progression throughout the paper, however some parts are dense and technical, which might be challenging for readers not deeply familiar with game theory in FL. More intuitive explanations or visual aids could enhance understanding. The inclusion of empirical results is commendable, but the paper could use more illustrative graphics or flowcharts to depict the workings of the PAGE algorithm.

3- Novelty/Quality of the Method
The concept of balancing personalisation and generalisation in FL using game theory is novel and well-articulated. The PAGE algorithm is the first to approach this problem from a game-theoretic perspective, filling a significant gap in FL research. The method is highly innovative, applying game theory to balance personalisation and generalisation in FL, a novel approach in this field. The paper presents a rigorous theoretical foundation for the PAGE algorithm. The use of Markov decision processes and reinforcement learning to simplify the equilibrium finding process is particularly noteworthy. However the claim at the end of Section 2, "no prior arts take the balance of local model personalization and global model generalization into account", appears to be incorrect due to the work shown in [9]. This method uses a two-predictor framework so explicitly stating that this is the "first algorithm" or "first application of game theory to this problem" would be better than the general statement above. While the theoretical foundation is robust, the paper could benefit from a more detailed explanation of the algorithm’s implementation.

4- Quality of the Evaluation
The experiments are well-designed, covering multiple datasets and comparing PAGE with 10 state-of-the-art FL baselines. The results, showing improvements of up to 35.20% and 39.91% in global and local prediction accuracy respectively, are impressive and lend strong support to the proposed approach. The description of the environment used for the experiment (Section 5), as well as the hyperparameters in Appendix C.2 (Table 7), are presented clearly and concisely enough for easy reproducibility. The results indicate a substantial improvement in both global and local prediction accuracy. However, additional experiments or case studies could further validate the robustness of the algorithm across various scenarios. The paper presents a solid statistical analysis of the results, but further statistical tests could be employed to strengthen the findings. It is unclear whether the FL settings described in 5.1 are tailored for the PAGE algorithm and the baselines wouldn’t be as optimised, so some clarity on this is needed. It is stated that the baselines are implemented with their original hyperparameters, however it is unclear whether there has been any experimenting to see if these are the optimal parameters for the specific FL settings used in the evaluation, which one can assume would’ve been done for the PAGE algorithm.
The interpretation of the results is mostly clear and aligns well with the paper’s objectives. However, a deeper discussion on why PAGE performs better under certain conditions would be beneficial. The results are contextualised within the current state of FL research, but the paper could benefit from a more detailed discussion on how these findings advance the field or could be applied in practical scenarios. While the paper briefly discusses future directions, a more comprehensive discussion on the limitations of the current study and potential areas for future research would enhance its impact. The paper presents results from specific datasets, but it lacks a discussion on the applicability of PAGE to a broader range of FL scenarios or datasets with varying characteristics. The paper does not adequately address potential limitations or challenges in implementing PAGE, such as computational overhead or scalability in large, real-world federated networks.

5- Overall
The paper presents a significant contribution to the field of Federated Learning with its novel game theory-based approach to optimisation. The strength of the theoretical foundation and the positive experimental results are commendable. To further enhance the paper, I recommend improvements in the clarity and readability of the methodology section, an expansion of the experimental scope, and a more detailed discussion of the practical implications and limitations of the PAGE algorithm (all of which are more thoroughly detailed above). With these modifications, the paper would be a strong candidate for acceptance.

**Questions:**

Please see my review above in addition to below.
1- Can you elaborate more on Fed-Rod drawbacks, can you motivate with empirical evidence?
2- Can you ensure the claim of being first to consider the global and personalization balance is valid
3- what is the proof that the gaming strategy is equivalent to an MDP?
4- Are you using average accuracy or something else and if so how is it calculated?
5- Section 4, is very hard to follow and read, can this be improved and if so how?
6- Can you provide a deeper discussion on why PAGE performs better under certain conditions would be beneficial?
7- Can you provide a more comprehensive discussion on the limitations of the current study and potential areas for future research that would enhance its impact?
8- How do the computational resources in Table 5 compare to other methods, without it, it is hard to judge?

**Reviewer Confidence:**

4: The reviewer is certain that the evaluation is correct and very familiar with the relevant literature

**Scope:**

3: The work is somewhat relevant to the Web and to the track, and is of narrow interest to a sub-community

---

### Official Review · Reviewer_krU1 · 2023-12-06

**Novelty:** 6
**Technical Quality:** 6

**Review:**

The paper introduces PAGE (Personalize and Generalize Equilibrium), a novel algorithm designed to address a crucial challenge in Federated Learning (FL): striking a balance between personalizing local models for individual clients and generalizing a global model applicable across all clients. This balance is essential in FL, where learning occurs across multiple decentralized devices or servers while maintaining data privacy. PAGE uniquely integrates game theory and reinforcement learning to dynamically adjust the balance based on specific learning scenarios and data distributions. The paper provides a comprehensive evaluation, comparing PAGE's performance with existing state-of-the-art FL methods across various datasets, highlighting its global and local prediction accuracy improvement.

Strengths
There are several key strengths of this paper:
1. The paper introduces a unique approach combining game theory and reinforcement learning. To my knowledge, this integration is not commonly seen in FL and represents an innovative way to tackle the balance between personalization and generalization in decentralized learning environments.

2. The algorithm effectively tackles one of the fundamental challenges in FL - balancing the need for personalized models for individual clients with a generalized model that performs well across all clients. This is a critical issue in FL, and the paper's approach to solving it is a significant contribution.

3. The paper provides a thorough and robust evaluation of the PAGE algorithm. It includes comparisons with state-of-the-art FL methods across various datasets, demonstrating global and local prediction accuracy improvements. This extensive testing validates the effectiveness of the algorithm.

Things to improve:
1. The paper could benefit from a more detailed exploration of the algorithm's scalability. Understanding how PAGE performs in larger, more diverse, real-world environments is crucial for assessing its practical utility.

2. A detailed discussion of the potential limitations of PAGE would be helpful for the readers. What are some of the shortcomings and risks associated with this new algorithm? Are there scenarios where its usefulness may be limited?

**Questions:**

I have mentioned two points of discussion above.

**Reviewer Confidence:**

2: The reviewer is willing to defend the evaluation, but it is likely that the reviewer did not understand parts of the paper

**Scope:**

4: The work is relevant to the Web and to the track, and is of broad interest to the community

---

### Decision · Program_Chairs · 2024-01-22

**Decision:**

Accept

**Comment:**

This paper presents an innovative approach for addressing a significant challenge in Federated Learning (FL), effectively balancing personalization and generalization. The methodological integration of game theory and reinforcement learning is particularly noteworthy. However, there are areas where the paper could be strengthened. These include a discussion of the algorithm's scalability, a comparison with similar works and clarifying some aspects of the paper.

 **Pros:**
 1. Innovative Approach: Integration of game theory and reinforcement learning in FL is novel.
 2. Effective Solution: The algorithm effectively balances personalization and generalization, a key challenge in FL.
 3. Sound Evaluation: Comprehensive testing against various datasets and state-of-the-art FL methods, demonstrating improvements in prediction accuracy.

 **Cons:**
 1. Scalability and Limitations: Lack of detailed exploration of the algorithm's scalability and potential limitations.
 2. Comparison with Similar Works: Comparison with related works or methods dealing with similar problems in FL needs improvement.
 3. Overheads: Lack of discussion about the computational overhead of current methods in comparison to the proposed method.
 4. Limited Discussion on Practical Implications: Insufficient discussion on how the findings advance FL research or can be applied in practical scenarios.
 5. Lack of Detailed Explanation: Certain algorithm details and decision-making processes need further clarification.